# Enhancing HIV positivity yield in southern Mozambique: The effect of a Ministry of Health training module in targeted provider-initiated testing and counselling

Anna Saura-Lázaro[1☯]*, Sheila Fernández-Luis[1,2☯], Tacilta Nhampossa[2,3], Laura Fuente-Soro[1,2], Elisa López-Varela[1,2], Edson Bernardo[2,4], Orvalho Augusto[2,5,6], Teresa Sánchez[7], Paula Vaz[8], Stanley C. Wei[9], Peter Kerndt[10], Nely Honwana[9], Peter Young[9], Guita Amane[11], Fernando Boene[11], Denise Naniche[1,2]

1 ISGlobal, Hospital Clínic, Universitat de Barcelona, Barcelona, Spain, 2 Centro de Investigação em Saúde de Manhiça (CISM), Maputo, Mozambique, 3 Instituto Nacional de Saúde (INS), Maputo, Mozambique, 4 Manhiça District Health Services, Maputo, Mozambique, 5 Faculdade de Medicina, Universidade Eduardo Mondlane, Maputo, Mozambique, 6 Department of Global Health, University of Washington, Seattle, Washington, United States of America, 7 London School of Hygiene and Tropical Medicine, London, United Kingdom, 8 Fundação Ariel Glaser Contra o SIDA Pediatrico, Maputo, Mozambique, 9 Division of Global HIV and Tuberculosis at Centers for Disease Control and Prevention (CDC), Maputo, Mozambique, 10 U.S Agency for International Development (USAID), Global Health, Washington, United States of America, 11 National STI-HIV/AIDS Programme, Ministry of Health, Maputo, Mozambique

☯ These authors contributed equally to this work.
* anna.saura@isglobal.org

## Abstract

In Mozambique, targeted provider-initiated HIV testing and counselling (PITC) is recommended where universal PITC is not feasible, but its effectiveness depends on healthcare providers' training. This study aimed to evaluate the effect of a Ministry of Health training module in targeted PITC on the HIV positivity yield, and identify factors associated with a positive HIV test. We conducted a single-group pre-post study between November 2018 and November 2019 in the triage and emergency departments of four healthcare facilities in Manhiça District, a resource-constrained semi-rural area. It consisted of two two-month phases split by a one-week targeted PITC training module ("observation phases"). The HIV positivity yield of targeted PITC was estimated as the proportion of HIV-positive individuals among those recommended for HIV testing by the provider. Additionally, we extracted aggregated health information system data over the four months preceding and following the observation phases to compare yield in real-world conditions ("routine phases"). Logistic regression analysis from observation phase data was conducted to identify factors associated with a positive HIV test. Among the 7,102 participants in the pre- and post-training observation phases (58.5% and 41.5% respectively), 68% were women, and 96% were recruited at triage. In the routine phases with 33,261 individuals (45.8% pre, 54.2% post), 64% were women, and 84% were seen at triage. While HIV positivity yield between pre- and post-training observation phases was similar (10.9% (269/2470) and 11.1% (207/1865), respectively), we observed an increase in yield in the post-training routine phase for women in triage, rising from 4.8% (74/1553) to 7.3% (61/831) (Yield ratio = 1.54; 95%CI: 1.11–

**Data Availability Statement:** The data cannot be publicly shared due to ethical restrictions. Data contain potentially sensitive information, and the national ethics committee (CNBS) does not authorize data sharing without a protocol request specifying the objectives and the researchers who will have access to the data. The datasets generated and/or analysed during the current study are available upon request (contact via llorenc. quinto@isglobal) for researchers who meet the criteria for accessing confidential data.

**Funding:** This publication has been supported by the President's Emergency Plan for AIDS Relief (PEPFAR) through the Centers for Disease Control and Prevention (CDC) under the terms of NU2GGH002092, the Severo Ochoa predoctoral fellowship by the Barcelona Institute of Global Health (ISGlobal) to ASL, the predoctoral fellowship from the Secretariat of Universities and Research, Ministry of Enterprise and Knowledge of the Government of Catalonia and cofounded by European Social Fund to ASL and SFL, and the European Respiratory Society (ERS) and the European Union (EU)'s H2020 research and innovation programme under the Marie Sklodowska-Curie grant agreement [847462] to ELV (This publication reflects only the author's view. The ERS, Research Executive Agency and EU are not responsible for any use that may be made of the information it contains). SW, employed by CDC participated in the conceptualization, study design and manuscript revision. For the remaining authors none were declared. The findings and conclusions in this report are those of the author(s) and do not necessarily represent the official position of the funding agencies.

**Competing interests:** The authors have declared that no competing interests exist.

2.14). Age (25–49 years) (OR = 2.43; 95%CI: 1.37–4.33), working in industry/mining (OR = 4.94; 95%CI: 2.17–11.23), unawareness of partner's HIV status (OR = 2.50; 95%CI: 1.91–3.27), and visiting a healer (OR = 1.74; 95%CI: 1.03–2.93) were factors associated with a positive HIV test. Including these factors in the targeted PITC algorithm could have increased new HIV diagnoses by 2.6%. In conclusion, providing refresher training and adapting the current targeted PITC algorithm through further research can help reach undiagnosed PLHIV, treat all, and ultimately eliminate HIV, especially in resource-limited rural areas.

## Introduction

The Joint United Nations Programme on HIV/AIDS (UNAIDS) set the "95-95-95" targets as part of the strategy to end the AIDS epidemic by 2030 [1]. These targets aim to ensure that by 2025, 95% of people living with HIV (PLHIV) know their HIV status, 95% of people diagnosed with HIV receive sustained antiretroviral therapy (ART), and 95% of those on ART achieve viral load suppression. The first 95 target is the single most important step as it enables access to treatment and care. In 2021, Eastern and Southern Africa remained the regions with the highest number of PLHIV, with 20.6 million ─54% of all PLHIV in the world [2]. While many countries of the region have achieved the second and third "95-95-95" targets, they still struggle with the first one [2]. By the end of 2021, the gap to achieve the first 95 was 1.3 million among all PLHIV in Eastern and Southern Africa. In Mozambique, one of the most severely affected countries in the world by the HIV epidemic, out of the 2.1 million PLHIV in 2021, 84% knew their status [3]. To address this first 95 target, the World Health Organization (WHO) recommends universal provider-initiated testing and counselling (PITC) for all individuals attending healthcare facilities in high HIV burden countries [4]. HIV positivity yield, defined as the proportion of individuals who newly test positive for HIV out of the total individuals tested, has been often used as an operational indicator to monitor the performance of HIV testing programmes [5, 6].

As countries approach the "95-95-95" targets, the number of undiagnosed PLHIV decreases, leading to fewer new HIV infections. However, this also means that more HIV tests are needed to detect the remaining undiagnosed PLHIV, resulting in a decline in HIV positivity yield [7]. This along with limited resources and capacity has led to consideration of alternative HIV testing and counselling (HTC) modalities focused on people at higher risk [8]. Targeted PITC, through which healthcare providers offer HTC to individuals presenting with risk factors, signs or symptoms suggestive of underlying HIV infection, has been shown to improve HIV positivity yield and efficiency, particularly in many Sub-Saharan African (SSA) countries [9–12]. In Mozambique, targeted PITC is the modality that identifies the largest number of new HIV diagnoses, especially among adults over 25 years, with the national HIV positivity yield in 2021 estimated at 6% [3, 13]. Prioritization for HIV testing through targeted PITC depends on healthcare provider experience, training, daily patient load, and worker's turnover. Despite this, Mozambique lacked a standardized training program in targeted PITC for healthcare providers. To address this gap and optimise targeted PITC, the Mozambican Ministry of Health (MoH) developed a training module for healthcare providers [14].

Our primary objective was to evaluate the effect of a new MoH training module in targeted PITC on the HIV positivity yield in the triage and emergency department (ED) in a rural area in southern Mozambique. As a secondary objective, we aimed to identify factors associated with a positive HIV test.

## Material and methods

### Study design, setting and population

We conducted a single-group pre-post study involving healthcare providers between November 2018 and November 2019 to evaluate the new MoH training module in targeted PITC as the intervention. The study was conducted in the four highest-volume healthcare facilities of Manhiça District (MD) —MD hospital, Xinavane rural hospital, and Palmeira and Maragra health units. MD is a semi-rural area in the Maputo province, southern Mozambique, characterized by a high community HIV prevalence of 36.6% in 2015, and the strain of its healthcare system, which operates with limited resources [15, 16]. It is served by one referral district hospital, one rural hospital and 15 peripheral health units which offer free HIV services. The Mozambican national HIV testing strategy recommends targeted PITC in the high-volume entry points of healthcare facilities where universal PITC is not feasible to be applied, such as the triage and ED [17, 18]. Triage is the first point of contact for clients who arrive at the health facility seeking medical attention. This department is responsible for rapidly assessing the severity of the client's condition and prioritizing them based on the urgency of their medical needs. If a client presents with severe or life-threatening conditions he/she is immediately referred to the ED.

The study consisted of two two-month individual-level data phases ("observation phases") split by a one-week targeted PITC training module: i) pre-training observation phase (1 March-12 May 2019), ii) training (13–19 May 2019), iii) post-training observation phase (20 May-31 July 2019) (Fig 1). We invited healthcare providers who worked routinely in the triage and ED to participate, and we include those who agreed to participate in both study observation phases and in the targeted PITC training module. Inclusion criteria for clients included being ≥15 years old, resident of MD, presenting to the triage or ED with non-disabling condition for participating in the study and giving informed consent (IC). We did not include those individuals who were <18 years old and came to the healthcare facility without a caregiver. Additionally, in order to control for the potential observer bias effect and estimate the yield in real-world conditions, we extracted data from the health information data system of all individuals presenting at triage and ED four months preceding and following both observation phases and training ("routine phases"): iv) pre-training routine phase (1 November 2018–28 February 2019), v) post-training routine phase (1 August-30 November 2019) (Fig 1). These data were aggregated by healthcare facility, department, day, sex and age group. These data were extracted in January 2020, and since they were aggregated, the authors could not identify individuals during or after data collection.

During the period before the intervention, comprising the pre-training routine and observation phases, healthcare providers followed the national targeted PITC algorithm for referring individuals for HIV testing. However, they had not received the new training module at that

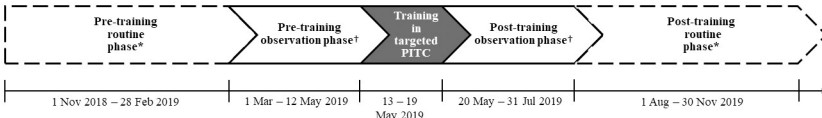

**Fig 1. Study timeline.** *Aggregated data extracted from the health information data system (real-world conditions). † Individual-level data collected through a study case report form (study conditions). Abbreviations: PITC: provider-initiated testing and counselling.

time. This contrasts with the period following the training, which included the post-training observation and routine phases.

## Targeted PITC algorithm and training module

The Mozambican national strategy for targeted PITC includes an algorithm based on risk factors, signs and symptoms to prioritize those individuals at higher risk for HIV testing (S1 Fig) [17]. An individual is eligible for HIV testing through targeted PITC when he/she reports at least one of the symptoms or risk factors included in the algorithm, does not have a previous HIV diagnosis and does not have a negative HIV test result in the previous three months.

The training module in targeted PITC, based on the algorithm, consisted of one week of sessions aimed at optimising healthcare providers' prioritization for HIV testing through targeted PITC. The module was designed and conducted in person by the HTC reference group of the Mozambican MoH at the MD hospital to ensure standardisation and reproducibility [14]. At the time of the study, the training module was in a pilot phase, and it was on the verge of being implemented. Since then it has become a standardised MoH-approved training module, but its routine implementation is not mandatory.

## Study visit procedures for observation phases

During the routine clinic visit, healthcare providers recorded their decision and reason to refer or not refer individuals for HIV testing in a paper-based case report form. Following the routine clinic visit with the provider, a study HIV counsellor invited the individuals who met the inclusion criteria to participate in the study. After obtaining IC, we offered HIV testing to the participant regardless of the provider's recommendation. This approach was intended to assess the number of new HIV diagnosis missed by targeted PITC and analyse the factors associated with a positive HIV test while minimising the potential selection bias. HIV testing was conducted following the Mozambican HIV testing algorithm which included two serial rapid diagnostic tests, Determine and Unigold [14]. We did not test those participants with a previously known HIV diagnosis. The study HIV counsellor also collected clinical information (risk factors, signs and symptoms of HIV infection according to the national targeted PITC algorithm) and sociodemographic characteristics through an electronic study questionnaire. We included additional factors that have been described in the literature to be associated with a positive HIV test [9]. We performed the same study visit procedures in both observation phases.

## Statistical analysis

We calculated proportions for categorical variables and the median and interquartile range (IQR) for continuous variables. Categorial variables were compared using Pearson's chi-squared test. We estimated HIV positivity yield (primary outcome) of targeted PITC as the proportion of HIV-positive individuals out of the total individuals referred by the provider and tested. We used the two-proportion z-test to test for the absolute difference in yield proportions of targeted PITC between the pre- and post-training observation phases and between the pre- and post-training routine phases, stratified by sex and health department. Moreover, we estimated yield ratios (YR) as a relative measure of associated gain using log-binomial regression models. We performed logistic regression analysis using individual-level data from the observation phases to assess the association of the signs and symptoms, risk factors and sociodemographic characteristics with a positive HIV test (secondary outcome). Participants with an undetermined result were excluded from this analysis. We built a multivariable model by including all variables with a p-value <0.20 in bivariable analyses (sex and age were fixed), followed by backward stepwise selection, removing variables with p-values >0.20 and adding

those with p-value <0.05. Furthermore, we stratified the participants enrolled during the observation phases by their eligibility for HIV testing under the current targeted PITC algorithm to estimate the number of new HIV diagnoses missed by the existing algorithm. We used Stata version 16 for the analyses [19].

### Ethics statement

This study was approved by the Mozambican MoH Institutional review board (IRB) (Approval Nr. 363/CNBS/18), the Centro de Investigação em Saúde de Mahiça IRB (Approval Nr. CIBS-- CISM/010/2018) and the Hospital Clinic Barcelona IRB (Approval Nr. HCB/2019/0379) and reviewed according to Centers for Disease Control and Prevention human research protection procedures (Approval Nr. 009F3EB81AA4251). All study participants, both healthcare providers and individuals presenting at the healthcare facilities, completed written IC. For participants aged 15–18 years, a parent/legal representative's additional consent was necessary. Regarding the routine phases, as we extracted de-identified aggregated data, informed consent was not required.

## Results

### Study profile for observation phases

A total of 19 healthcare providers, all medical assistants, participated in the study (three and six in the triage and ED of the MD hospital, respectively; two and two in the triage and ED of the Xinavane rural hospital, respectively; and three in the triage department of each of the Palmeira and Maragra health units). These participants accounted for 82% of the healthcare providers working across the four facilities. Out of the 8,533 clients screened for the study, we enrolled 83.2% as study participants —4,155 (58.5%) and 2,947 (41.5%) in the pre- and post-training observation phase, respectively— hereinafter referred to as client participants. Among them, 43.5% (n = 1,809) and 45.7% (n = 1,347) were eligible for HIV testing referral through targeted PITC in the pre- and post-training observation phases, respectively (Fig 2).

Healthcare providers referred 95.8% (1,733/1,809) of the algorithm-eligible individuals in the pre-training observation phase and 99.3% (1,338/1,347) in the post-training one. They also referred a third of algorithm-non-eligible individuals in the pre- and post-training observation phases: 35.9% (843/2,346) and 36.2% (579/1,600), respectively. Study HIV counsellors tested a total of 5,028 individuals without an already known HIV diagnosis, of whom 86.2% (n = 4,335) were referrals from providers (Fig 2).

Providers stated two main reasons for not recommending for HIV testing: an already known HIV diagnosis (n = 984; 62.3% and n = 684; 66.4% in the pre- and post-training phases, respectively) and a negative HIV test in the previous three months (n = 384; 24.3% and n = 269; 26.1% in the pre- and post-training phases, respectively). Not presenting with HIV risk symptoms was only reported as a reason for not referring to HIV testing among 6.5% (n = 101) and 1.1% (n = 11) in both respective phases.

### Client characteristics

Most of the client participants were women (n = 4,798; 68%), had a median age of 32 years (IQR: 23–45), and were recruited in the triage (n = 6,792; 96%) and in the MD and Xinavane hospitals (n = 2,565; 36% and n = 2,110; 30%, respectively). Slightly more women than men (42.5% vs. 39.5%, p-value = 0.015) and more individuals from the Xinavane rural hospital compared to the Maragra health unit (45.6% vs. 36.4, p-value<0.001) were recruited during the post-training observation phase (Table 1A).

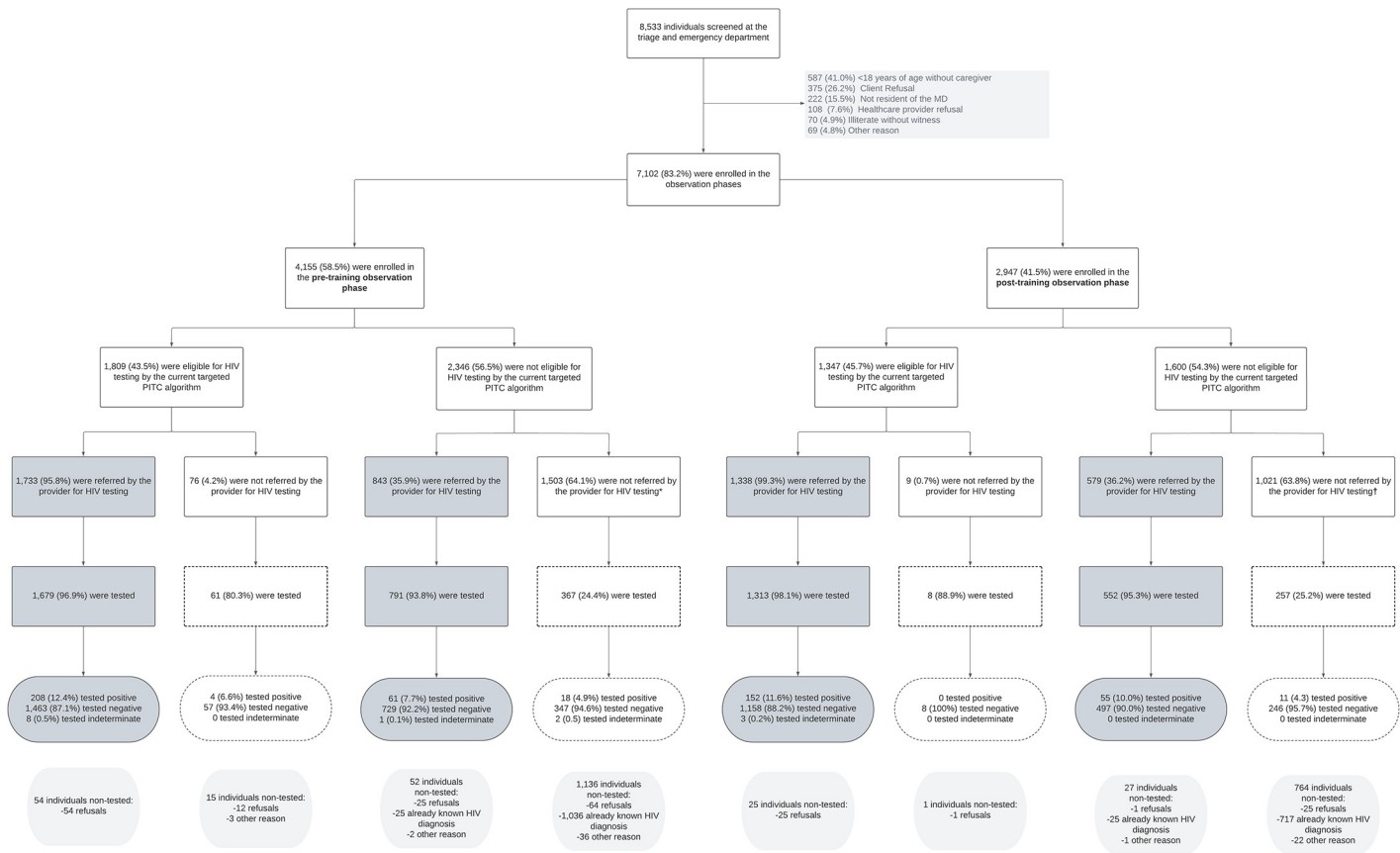

**Fig 2. Study profile of the clients enrolled in the observation phases.** * Including 384 individuals with a negative HIV test in the previous three months. † 367 individuals with a negative HIV test in the previous three months. ‡ Including 269 individuals with a negative HIV test in the previous three months. § 257 individuals with a negative HIV test in the previous three months. We defined eligible for HIV testing, according to the Mozambican national targeted PITC algorithm, as presenting with at least one HIV risk factor, sign or symptom and without an already known HIV diagnosis or negative HIV test in the previous three months. The areas shaded in dark grey correspond to the individuals tested by the targeted PITC modality for HIV testing and counselling. The boxes with dashed-lines correspond to individuals who were not tested by the targeted PITC modality, but who were tested because of study purposes. Study HIV counsellors tested all the individuals who did not have an already known HIV diagnosis. Abbreviations: PITC: provider-initiated testing and counselling.

When comparing characteristics from the pre- and post-training routine phases, we found that the proportion of women presenting at the healthcare facilities during the post-training phase was higher than men (56.3% vs. 50.7%, p<0.001), as well as, the proportion of individuals 25–49 years old compared to individuals 20–24 years old (57.8% vs. 48.8%; p<0.001) (Table 1B).

## HIV positivity yield

The HIV positivity yield of targeted PITC between the pre- and post-training observation phases was similar: 10.9% (269/2470); 95% confidence interval (CI): 9.7–12.2 and 11.1% (207/1865); 95%CI: 9.7–12.5, respectively (p-value = 0.828). However, when we compared the yield between the pre- and post-training routine phases, the yield rose from 6.1% (165/2709); 95% CI: 5.2–7.0 to 8.9% (146/1637); 95%CI: 7.5–10.3 (p-value = 0.001), representing a 46% increase in yield (YR = 1.46; 95%CI: 1.18–1.81) (Table 2A).

When stratifying by health department, the HIV positivity yield in the triage rose from 5.8% (144/2464); 95%CI: 4.9–6.8 in the pre-training routine phase to 8.3% (112/1344); 95%CI:

**Table 1. Client characteristics at enrolment.**

A) Pre- and post-training observation phases (n = 7102)[a]

| | | N (%) | | p-value |
|---|---|---|---|---|
| | Total (n = 7102) | Pre-training observation phase (1 Mar –12 May 2019) (n = 4155) | Post-training observation phase (20 May– 31 Jul 2019) (n = 2947) | |
| **Age groups (in years)** | | | | |
| **15–19** | 619 (8.72) | 358 (57.84) | 261 (42.16) | 0.766[b] |
| **20–24** | 1417 (19.95) | 846 (59.70) | 571 (40.30) | |
| **25–49** | 3723 (52.42) | 2165 (58.15) | 1558 (41.85) | |
| **≥50** | 1343 (18.91) | 786 (58.53) | 557 (41.47) | |
| **Sex** | | | | **0.015**[b] |
| **Male** | 2304 (32.44) | 1395 (60.55) | 909 (39.45) | |
| **Female** | 4798 (67.56) | 2760 (57.52) | 2038 (42.48) | |
| **Healthcare facility** | | | | **<0.001**[b] |
| **Manhiça district hospital** | 2565 (36.12) | 1516 (59.10) | 1049 (40.90) | |
| **Xinavane rural hospital** | 2110 (29.71) | 1148 (54.41) | 962 (45.59) | |
| **Palmeira health unit** | 1049 (14.77) | 615 (58.63) | 434 (41.37) | |
| **Maragra health unit** | 1378 (19.4) | 876 (63.57) | 502 (36.43) | |
| **Department** | | | | 0.940[b] |
| **Triage** | 6792 (95.64) | 3973 (58.50) | 2819 (41.50) | |
| **Emergency** | 310 (4.36) | 182 (58.71) | 128 (41.29) | |
| **Occupation** | | | | |
| **Industry/Mining** | 35 (0.49) | 17 (48.57) | 18 (51.43) | **0.038**[b] |
| **Agriculture** | 1616 (22.75) | 929 (57.49) | 687 (42.51) | |
| **Own or employer's business** | 419 (5.90) | 226 (53.94) | 193 (46.06) | |
| **Street vendor** | 120 (1.69) | 79 (65.83) | 41 (34.17) | |
| **Student** | 569 (8.01) | 341 (59.93) | 228 (40.07) | |
| **Construction-related work** | 207 (2.91) | 137 (66.18) | 70 (33.82) | |
| **State worker** | 178 (2.51) | 98 (55.06) | 80 (44.94) | |
| **Unemployed** | 1999 (28.15) | 1180 (59.03) | 819 (40.97) | |
| **Other** | 151 (2.13) | 81 (53.64) | 70 (46.36) | |
| **Missing** | 1808 (25.46) | 1067 (59.02) | 741 (40.98) | |

B) Pre- and post-training routine phases (n = 33261)[c]

| | | N (%) | | p-value |
|---|---|---|---|---|
| | Total (n = 33261) | Pre-training routine phase (1 Nov 2018–28 Feb 2019) (n = 15220) | Post-training routine phase (1 Aug– 30 Nov 2019) (n = 18041) | |
| **Age groups (in years)** | | | | |
| **15–19** | 5456 (16.40) | 2584 (47.36) | 2872 (52.64) | **<0.001**[b] |
| **20–24** | 6098 (18.33) | 3123 (51.21) | 2975 (48.79) | |
| **25–49** | 14544 (43.73) | 6136 (42.19) | 8408 (57.81) | |
| **≥50** | 7163 (21.54) | 3377 (47.15) | 3786 (52.85) | |
| **Sex** | | | | **<0.001**[b] |
| **Male** | 12108 (36.40) | 5970 (49.31) | 6138 (50.69) | |
| **Female** | 21153 (63.60) | 9250 (43.73) | 11903 (56.27) | |

(*Continued*)

**Table 1.** (Continued)

| Department | | | | 0.211[b] |
|---|---|---|---|---|
| Triage | 27939 (84.00) | 12743 (45.61) | 15196 (54.39) | |
| Emergency | 5322 (16.00) | 2477 (46.54) | 2845 (53.46) | |

The table uses column proportions in the column "Total" and row proportions between columns "Pre-training observation/routine phase" and "Post-training observation/routine phase"

[a] During the observation phases, we recruited individuals attending the healthcare facility between 8 AM to 3 PM, Monday to Friday.

[b] Pearson's chi-squared test

[c] During the routine phases, we extracted aggregated data from data collection books that included all individuals accessing healthcare facilities round the clock, including weekends and bank holidays.

Abbreviations: IQR: interquartile range

6.9–9.8 in the post-training one (p-value = 0.003). This represented a 43% increase in yield in triage in the months following the training and the observation phases (YR = 1.43; 95%CI: 1.12–1.81). However, there was no significant increase in yield between the pre- and post-training routine phases in the ED (Table 2B).

Furthermore, when stratifying by sex and department, we found that the increase in yield in the triage during the post-training routine phase was significant among women (YR = 1.54; 95%CI: 1.11–2.14) and not men (YR = 1.29; 95%CI: 0.92–1.83). However, we observed greater yield values for men than for women in both the pre- and post-training routine phases (7.7% (70/911); 95%CI: 6.0–9.4 vs. 4.8% (74/1553); 95%CI: 3.7–5.8, p-value = 0.020 and 9.9% (51/513); 95%CI: 7.4–12.3 vs. 7.3% (61/831); 95%CI: 5.6–9.1, p-value = 0.007, respectively) (Table 2C and 2D).

## Factors associated with a positive HIV test

Among the 7102 participants in the observation phases, 5028 were tested, resulting in 509 (10.1%) positive HIV tests and 14 undetermined results. HIV prevalence differed by sex (9.4% among women compared to 11.4% among men, p-value = 0.025) and health department (9.7% in triage compared to 19.6% in the ED, p-value<0.001) (Table 3).

Multivariable analysis revealed that most of the variables included in the existing national targeted PITC algorithm were significantly associated with a positive HIV test. Among the other variables analysed, 25–49 years of age (OR = 2.43; 95%CI: 1.37–4.33), presenting at the MD district hospital (OR = 1.78; 95%CI: 1.33–2.39) and working in the industry or mining (OR = 4.94; 95%CI: 2.17–11.23) were the sociodemographic factors associated with a positive HIV test. Regarding risk factors, unawareness of partner's HIV status (OR = 2.50 (95%CI; 1.91–3.27) and having visited a healer in the previous six months (OR = 1.74; 95%CI: 1.03–2.93) were risk factors not included in the national PITC algorithm that showed a significant association with a positive HIV test. Likewise, presenting at the ED (OR = 1.75; 95%CI: 1.12–2.72), unawareness of pregnancy status among women (OR = 2.23; 95%CI: 1.01–4.94, (S1 Table)) and reporting night sweats (OR = 15.26; 95%CI: 2.53–91.97) were other factors significantly associated with a positive HIV result (Table 3).

Among all participants who tested positive for HIV, 72.0% (364 out of 509) were eligible for testing under the current targeted PITC algorithm. This leaves 145 HIV diagnoses that were missed by the existing algorithm. Regarding the factors not currently covered by the algorithm, there was no client unaware of the pregnancy status or who reported night

**Table 2. HIV positivity yield of targeted PITC by study phase, health department and sex.** Yield ratios from log-binomial regression model.

| | A) OVERALL | | | | | |
|---|---|---|---|---|---|---|
| Period and data source | Client participants / individuals presenting (N)[a] | Individuals referred and tested (N) | HIV-positive individuals (N) | HIV positivity yield (%) (95%CI) | Yield ratio (95% CI) | p-value[b] |
| **Pre-training observation phase** (1 Mar—12 May 2019) | 4155 | 2470 | 269 | 10.89 (9.66–12.11) | 1.02 (0.86–1.21) | 0.828 |
| **Post-training observation phase** (20 May—31 Jul 2019) | 2947 | 1865 | 207 | 11.10 (9.67–12.52) | | |
| **Pre-training routine phase** (1 Nov 2018–28 Feb 2019) | 15220 | 2709 | 165 | 6.09 (5.19–6.99) | 1.46 (1.18–1.81) | **0.001** |
| **Post-training routine phase** (1 Aug- 30 Nov 2019) | 18041 | 1637 | 146 | 8.92 (7.54–10.30) | | |

| B) ALL | TRIAGE | | | | | | EMERGENCY | | | | | |
|---|---|---|---|---|---|---|---|---|---|---|---|---|
| Period and data source | Client participants / individuals presenting (N)[a] | Individuals referred and tested (N) | HIV-positive individuals (N) | HIV positivity yield (%) (95% CI) | Yield ratio (95% CI) | p-value[b] | Client participants / individuals presenting (N)[a] | Individuals referred and tested (N) | HIV-positive individuals (N) | HIV positivity yield (%) (95% CI) | Yield ratio (95% CI) | p-value[b] |
| **Pre-training observation phase** (1 Mar—12 May 2019) | 3973 | 2330 | 236 | 10.13 (8.90–11.35) | 1.07 (0.89–1.28) | 0.484 | 182 | 140 | 33 | 23.57 (16.54–30.60) | 0.70 (0.41–1.20) | 0.186 |
| **Post-training observation phase** (20 May—31 Jul 2019) | 2819 | 1768 | 191 | 10.80 (9.36–12.25) | | | 128 | 97 | 16 | 16.49 (9.11–23.88) | | |
| **Pre-training routine phase** (1 Nov 2018–28 Feb 2019) | 12743 | 2464 | 144 | 5.84 (4.92–6.77) | 1.43 (1.12–1.81) | **0.003** | 2477 | 245 | 21 | 8.57 (5.07–12.08) | 1.10 (0.61–1.97) | 0.748 |
| **Post-training routine phase** (1 Aug—30 Nov 2019) | 15196 | 1344[c] | 112[c] | 8.33 (6.86–9.81) | | | 2845 | 212[c] | 20[c] | 9.43 (5.50–13.37) | | |

| C) MEN | TRIAGE | | | | | | EMERGENCY | | | | | |
|---|---|---|---|---|---|---|---|---|---|---|---|---|
| Period and data source | Client participants / individuals presenting (N)[a] | Individuals referred and tested (N) | HIV-positive individuals (N) | HIV positivity yield (%) (95% CI) | Yield ratio (95% CI) | p-value[b] | Client participants / individuals presenting (N)[a] | Individuals referred and tested (N) | HIV-positive individuals (N) | HIV positivity yield (%) (95% CI) | Yield ratio (95% CI) | p-value[b] |

(*Continued*)

2
2
1
2
2
1
2
1
2
2
1
2
2
2
2
1
2
1
2
2
1

**Table 2.** (Continued)

| | | | | | | | | | | | |
|---|---|---|---|---|---|---|---|---|---|---|---|
| **Pre-training observation phase** (1 Mar—12 May 2019) | 1315 | 922 | 97 | 10.52 (8.54–12.50) | 1.22 (0.92–1.61) | 0.164 | 80 | 61 | 13 | 21.31 (11.03–31.59) | 0.94 (0.44–2.00) | 0.869 |
| **Post-training observation phase** (20 May—31 Jul 2019) | 856 | 616 | 79 | 12.82 (10.18–15.47) | | | 53 | 45 | 9 | 20.00 (8.31–31.69) | | |
| **Pre-training routine phase** (1 Nov 2018–28 Feb 2019) | 4898 | 911 | 70 | 7.68 (5.95–9.41) | 1.29 (0.92–1.83) | 0.142 | 1072 | 123 | 7 | 5.69 (1.60–9.79) | 1.79 (0.71–4.53) | 0.218 |
| **Post-training routine phase** (1 Aug—30 Nov 2019) | 4903 | 513[c] | 51[c] | 9.94 (7.35–12.25) | | | 1235 | 98[c] | 10[c] | 10.20 (4.21–16.20) | | |

| **D) WOMEN** | TRIAGE | | | | | | EMERGENCY | | | | | |
|---|---|---|---|---|---|---|---|---|---|---|---|---|
| **Period and data source** | **Client participants / individuals presenting (N)[a]** | **Individuals referred and tested (N)** | **HIV-positive individuals (N)** | **HIV positivity yield (%) (95% CI)** | **Yield ratio (95% CI)** | **p-value[b]** | **Client participants / individuals presenting (N)[a]** | **Individuals referred and tested (N)** | **HIV-positive individuals (N)** | **HIV positivity yield (%) (95% CI)** | **Yield ratio (95% CI)** | **p-value[b]** |
| **Pre-training observation phase** (1 Mar—12 May 2019) | 2658 | 1408 | 139 | 9.87 (8.31–11.43) | 0.98 (0.78–1.25) | 0.899 | 102 | 79 | 20 | 25.32 (15.73–34.90) | 0.53 (0.24–1.17) | 0.115 |
| **Post-training observation phase** (20 May- 31 Jul 2019) | 1963 | 1152 | 112 | 9.72 (8.01–11.43) | | | 75 | 52 | 7 | 13.46 (4.18–22.74) | | |
| **Pre-training routine phase** (1 Nov 2018–28 Feb 2019) | 7845 | 1553 | 74 | 4.76 (3.71–5.82) | 1.54 (1.11–2.14) | **0.010** | 1405 | 122 | 14 | 11.48 (5.82–17.13) | 0.76 (0.35–1.65) | 0.494 |
| **Post-training routine phase** (1 Aug—30 Nov 2019) | 10293 | 831[c] | 61[c] | 7.34 (5.57–9.11) | | | 1610 | 114[c] | 10[c] | 8.77 (3.58–13.96) | | |

We performed the training module in targeted PITC from 13 to 19 May 2019.

[a] Client participants for the pre- and post-training observation phases (study conditions; individual-level data collected through a study case report form). Individuals presenting for the pre- and post-training routine phases (real-world conditions; aggregated data extracted from the health information data system).

[b] Two-proportion z-test

[c] Department is missing among 81 individuals referred and tested and 14 HIV-positive individuals, both in the post-training routine phase.

Abbreviations: CI: confidence interval

**Table 3. Factors associated with a positive HIV test.** Crude and adjusted odds ratios from logistic regression analyses.

| | Total (n = 5014)[b] | Positive HIV test (n = 509) | cOR | 95% CI | p-value | aOR[a] | 95% CI | p-value |
|---|---|---|---|---|---|---|---|---|
| **Sociodemographic characteristics** | | | | | | | | |
| **Age group (in years)[c]** | | | | | | | | |
| 15–19 | 552 (11.01) | 20 (3.62) | Ref. | | | Ref. | | |
| 20–24 | 1192 (23.77) | 93 (7.80) | 2.25 | 1.37–3.69 | **0.001** | 1.49 | 0.83–2.68 | 0.179 |
| 25–49 | 2313 (46.13) | 344 (14.87) | 4.64 | 2.93–7.37 | **0.000** | 2.43 | 1.37–4.33 | **0.002** |
| ≥50 | 957 (19.09) | 52 (5.43) | 1.53 | 0.90–2.59 | 0.114 | 0.74 | 0.37–1.48 | 0.391 |
| **Sex[c]** | | | | | | | | |
| Female | 3211 (64.04) | 303 (9.44) | Ref. | | | Ref. | | |
| Male | 1803 (35.96) | 206 (11.43) | 1.24 | 1.03–1.49 | **0.025** | 1.13 | 0.86–1.48 | 0.392 |
| **Healthcare facility[c]** | | | | | | | | |
| Xinavane rural hospital | 1579 (31.49) | 126 (7.98) | Ref. | | | Ref. | | |
| Manhiça district hospital | 1864 (37.18) | 235 (12.61) | 1.66 | 1.32–2.09 | **<0.001** | 1.78 | 1.33–2.39 | **<0.001** |
| Palmeira health unit | 683 (13.62) | 79 (11.57) | 1.51 | 1.12–2.03 | **0.007** | 0.96 | 0.64–1.45 | 0.848 |
| Maragra health unit | 888 (17.71) | 69 (7.77) | 0.97 | 0.72–1.32 | 0.853 | 0.67 | 0.44–1.03 | 0.070 |
| **Occupation[c]** | | | | | | | | |
| Farmer | 1544 (30.79) | 142 (9.20) | Ref. | | | Ref. | | |
| Industry/Miner | 34 (0.68) | 13 (38.24) | 6.11 | 3.00–12.47 | **<0.001** | 4.94 | 2.17–11.23 | **<0.001** |
| Own or employer's business | 399 (7.96) | 65 (16.29) | 1.92 | 1.40–2.64 | **<0.001** | 1.29 | 0.87–1.90 | 0.202 |
| Street vendor | 113 (2.25) | 15 (13.27) | 1.51 | 0.86–2.67 | 0.156 | 1.01 | 0.52–1.97 | 0.974 |
| Student | 535 (10.67) | 20 (3.74) | 0.38 | 0.24–0.62 | **<0.001** | 0.61 | 0.32–1.15 | 0.126 |
| Construction-related work | 199 (3.97) | 32 (16.08) | 1.89 | 1.25–2.87 | **0.003** | 1.22 | 0.73–2.05 | 0.451 |
| State worker | 166 (3.31) | 16 (9.64) | 1.05 | 0.61–1.81 | 0.852 | 0.54 | 0.27–1.08 | 0.079 |
| Unemployed | 1874 (37.38) | 185 (9.87) | 1.08 | 0.86–1.36 | 0.504 | 1.10 | 0.80–1.48 | 0.583 |
| Other | 147 (2.93) | 20 (13.61) | 1.56 | 0.94–2.57 | 0.085 | 1.17 | 0.64–2.15 | 0.605 |
| Missing | 3 (0.06) | 1 (33.33) | NA | | | | | |
| **Risk factors** | | | | | | | | |
| **HIV-positive partner (n = 3965[d])** | | | | | | | | |
| No | 2555 (64.43) | 182 (7.12) | Ref. | | | Ref. | | |
| Yes | 231 (5.83) | 64 (27.71) | 5.00 | 3.61–6.92 | **<0.001** | 4.84 | 3.39–6.89 | **<0.001** |
| Don't know[c] | 1168 (29.46) | 160 (13.70) | 2.07 | 1.65–2.59 | **<0.001** | 2.50 | 1.91–3.27 | **<0.001** |
| Missing | 11 (0.28) | 3 (27.70) | NA | | | | | |
| **More than one sexual partner in the past year (n = 3965[d])** | | | | | | | | |
| No | 3655 (92.18) | 342 (9.36) | Ref. | | | Ref. | | |
| Yes | 282 (7.11) | 63 (22.34) | 2.79 | 2.06–3.77 | **<0.001** | 1.70 | 1.18–2.47 | **0.005** |
| Missing | 28 (0.71) | 4 (14.29) | NA | | | | | |
| **Condom use (n = 3965[d])** | | | | | | | | |
| Never | 1760 (44.38) | 215 (12.22) | Ref. | | | Ref. | | |
| Occasionally[e] | 1351 (34.07) | 129 (9.55) | 0.76 | 0.60–0.96 | **0.019** | 0.79 | 0.60–1.04 | **0.091** |
| Frequently[e] | 442 (11.15) | 36 (8.14) | 0.64 | 0.44–0.92 | **0.017** | 0.65 | 0.43–0.97 | **0.036** |
| Always | 386 (9.74) | 25 (6.48) | 0.50 | 0.32–0.76 | **0.001** | 0.52 | 0.33–0.84 | **0.008** |
| Missing | 26 (0.66) | 4 (15.38) | NA | | | | | |
| **Police arrest[c]** | | | | | | | | |
| No | 4936 (98.44) | 493 (9.99) | Ref. | | | Ref. | | |
| Yes, last year | 30 (0.60) | 7 (23.33) | 2.74 | 1.17–6.42 | **0.020** | | | |

*(Continued)*

**Table 3.** (*Continued*)

| | Total (n = 5014)[b] | Positive HIV test (n = 509) | cOR | 95% CI | p-value | aOR[a] | 95% CI | p-value |
|---|---|---|---|---|---|---|---|---|
| Yes, more than a year ago | 41 (0.82) | 8 (19.51) | 2.19 | 1.00–4.76 | **0.049** | | | |
| Missing | 7 (0.14) | 1 (14.29) | NA | | | | | |
| **Hospitalised in the previous 6 months** | | | | | | | | |
| No | 4853 (96.79) | 480 (9.89) | Ref. | | | | | |
| Yes | 148 (2.95) | 27 (18.24) | 2.03 | 1.33–3.12 | **0.001** | | | |
| Missing | 13 (0.26) | 2 (15.39) | NA | | | | | |
| **Visited a healer in the previous 6 months**[c] | | | | | | | | |
| No | 4852 (96.77) | 479 (9.87) | Ref. | | | Ref. | | |
| Yes | 145 (2.89) | 27 (18.62) | 2.09 | 1.36–3.21 | **0.001** | 1.74 | 1.03–2.93 | **0.039** |
| Missing | 17 (0.34) | 3 (17.65) | NA | | | | | |
| **Other factors** | | | | | | | | |
| **Health department**[c] | | | | | | | | |
| Triage | 4764 (95.01) | 460 (9.66) | Ref. | | | Ref. | | |
| Emergency | 250 (4.99) | 49 (19.60) | 2.28 | 1.65–3.16 | **<0.001** | 1.75 | 1.12–2.72 | **0.014** |
| **Pregnant or partner of pregnant woman**[c, f] | | | | | | | | |
| No | 4818 (96.09) | 474 (9.84) | Ref. | | | Ref. | | |
| Yes | 131 (2.61) | 23 (17.56) | 1.95 | 1.23–3.09 | **0.004** | 1.42 | 0.85–2.39 | 0.178 |
| Don't know | 58 (1.16) | 11 (18.97) | 2.15 | 1.11–4.16 | **0.024** | 2.68 | 1.31–5.45 | **0.007** |
| Missing | 7 (0.14) | 1 (14.29) | NA | | | | | |
| **Signs and symptoms** | | | | | | | | |
| **Headache**[c] | | | | | | | | |
| No | 4766 (95.05) | 469 (9.84) | Ref. | | | | | |
| Yes | 248 (4.95) | 40 (16.13) | 1.76 | 1.24–2.50 | **0.002** | | | |
| **Diarrhoea/Vomiting/Abdominal pain** | | | | | | | | |
| No | 4900 (97.73) | 490 (10.00) | Ref. | | | | | |
| Yes | 114 (2.27) | 19 (16.67) | 1.8 | 1.09–2.97 | **0.022** | | | |
| **Skin or oral mucosa lesions** | | | | | | | | |
| No | 4927 (98.26) | 485 (9.84) | Ref. | | | Ref. | | |
| Yes | 87 (1.74) | 24 (27.59) | 3.49 | 2.16–5.64 | **<0.001** | 3.96 | 2.19–7.18 | **<0.001** |
| **Genital wounds or discharge** | | | | | | | | |
| No | 4852 (96.77) | 493 (10.16) | Ref. | | | | | |
| Yes | 162 (3.23) | 16 (9.88) | 0.97 | 2.57–1.64 | 0.906 | | | |
| **Cough for over 3 weeks** | | | | | | | | |
| No | 4910 (97.93) | 481 (9.80) | Ref. | | | Ref. | | |
| Yes | 104 (2.07) | 28 (26.92) | 3.39 | 2.18–5.29 | **<0.001** | 1.87 | 1.03–3.41 | **0.041** |
| **Night sweats**[c] | | | | | | | | |
| No | 5002 (99.76) | 500 (10.00) | Ref. | | | Ref. | | |
| Yes | 12 (0.24) | 9 (75.00) | 27.01 | 7.29–100.10 | **<0.001** | 15.26 | 2.53–91.97 | **0.003** |
| **Sore throat**[c] | | | | | | | | |
| No | 4986 (99.44) | 504 (10.11) | Ref. | | | | | |
| Yes | 28 (0.56) | 5 (17.86) | 1.93 | 0.73–5.11 | 0.184 | | | |
| **Fever for over 7 days** | | | | | | | | |
| No | 4994 (99.60) | 502 (10.05) | Ref. | | | | | |
| Yes | 20 (0.40) | 7 (35.00) | 4.82 | 1.91–12.13 | **0.001** | | | |

(*Continued*)

**Table 3.** (Continued)

| | | | cOR | 95% CI | p-value | aOR[a] | 95% CI | p-value |
|---|---|---|---|---|---|---|---|---|
| | Total (n = 5014)[b] | Positive HIV test (n = 509) | | | | | | |
| **Fever for less than 7 days[c]** | | | | | | | | |
| No | 4971 (99.14) | 502 (10.10) | Ref. | | | | | |
| Yes | 43 (0.86) | 7 (16.28) | 1.73 | 0.77–3.91 | 0.187 | | | |
| **Constitutional syndrome (asthenia anorexia weight loss)** | | | | | | | | |
| No | 4981 (99.34) | 498 (10.00) | Ref. | | | Ref. | | |
| Yes | 33 (0.66) | 11 (33.33) | 4.50 | 2.17–9.34 | **<0.001** | 2.42 | 0.91–6.44 | 0.077 |
| **Joint pain[c]** | | | | | | | | |
| No | 4858 (96.89) | 490 (10.09) | Ref. | | | | | |
| Yes | 156 (3.11) | 19 (12.18) | 1.24 | 0.76–2.02 | 0.395 | | | |
| **Difficulties in breathing** | | | | | | | | |
| No | 5002 (99.76) | 506 (10.12) | Ref. | | | | | |
| Yes | 12 (0.24) | 3 (25.00) | 2.96 | 0.80–10.98 | 0.104 | | | |
| **Urinary symptoms[c]** | | | | | | | | |
| No | 4973 (99.18) | 502 (10.09) | Ref. | | | | | |
| Yes | 41 (0.82) | 7 (17.07) | 1.83 | 0.81–4.16 | 0.147 | | | |
| **Other** | | | | | | | | |
| No | 4959 (98.90) | 503 (10.14) | Ref. | | | | | |
| Yes | 55 (1.10) | 6 (10.91) | 1.09 | 0.46–2.55 | 0.852 | | | |

The table uses column proportions in the column "Total" and row proportions in the column "Positive HIV test".

[a] We included 3893 participants in the multivariable analysis. We built a multivariable logistic regression model by including all variables with a p-value <0.20 in the bivariable analyses (sex and age were fixed), followed by backward stepwise selection, where variables with p-values <0.05 could enter the model whereas a p-value <0.20 was required to be retained. The final multivariable model excluded police arrest, being hospitalised, headache, diarrhoea/vomiting/abdominal pain and fever for over seven days.

[b] Client participants tested for HIV in both observation phases. 14 individuals with an undetermined HIV test result were excluded.

[c] Sociodemographic characteristics, risk factors or signs and symptoms not included in the national targeted PITC algorithm of Mozambique.

[d] 1049 participants reported not having any sexual partner.

[e] Occasionally: less than half of the times; Frequently: more than half of the times

[f] When performing the multivariable analysis stratified by sex being unaware of the pregnancy status was only significant among women (OR = 2.23; 95%CI: 1.01–4.94) (S1 Table).

Abbreviations: aOR: adjusted odds ratio, CI: confidence interval, cOR: crude odds ratio, NA: not applicable, PITC: provider-initiated testing and counselling, Ref.: reference category

sweats and who was not identified through the current algorithm as they presented at least one of the factors already included. In contrast, with the addition of age group, unawareness of partner's HIV status, having visited a healer during the previous six months and working in industry or mining to the PITC algorithm, providers might have been able to reach 13 more undiagnosed PLHIV who were not eligible for HIV testing by the current targeted PITC algorithm. These individuals represent 2.6% (13/509) out of the individuals with a positive HIV test. Furthermore, testing individuals with at least one risk factor or symptom but a recent negative HIV test revealed 18 HIV-positive individuals. The current national targeted PITC algorithm excludes such individuals, but the removal of this criterion could have increased new HIV diagnoses through targeted PITC by 3.5% (18/509).

## Discussion

This study found that the HIV positivity yield of targeted PITC increased by approximately 50% after the administration of a MoH training module to healthcare providers in the triage of four high-volume healthcare facilities in the semi-rural area of MD. This increase was observed in the post-training routine phase and specifically among women, while men had higher HIV positivity yield values but showed no increase in the post-training phase. No differences were observed between the pre- and post-training observation phases. Additionally, we identified that age (25–49 years), occupation (industry or mining), unawareness of partner's HIV status and a recent visit to a healer were factors associated with a positive HIV test which are not included in the current targeted PITC algorithm.

During both observation phases, healthcare providers referred most of the targeted PITC algorithm-eligible individuals, but they also referred those individuals who did not present symptoms or risk factors for HIV and did not have an already known HIV diagnosis or a recent HIV negative test. The study itself is likely to have influenced providers to refer in a manner closer to universal PITC than to targeted PITC, even before they received training, due to their awareness of the ongoing evaluation of the PITC performance. This is known as the Hawthorne effect, which has been described in several SSA healthcare settings [20–23]. Additionally, resources such as the number of HIV counsellors were increased for study purposes, so providers knew that all individuals referred for testing during the observation phases would be tested, in contrast to routine conditions. These factors may explain why there was not an increase in HIV positivity yield following the training module during the observation phase.

In real-world routine conditions, HIV positivity yield among women presenting at triage was 54% higher in the post-training phase compared to the pre-training one. This suggests that the training module in targeted PITC may successfully increase HIV positivity yield, as these differences were revealed when the study conditions—Hawthorne effect and extra HIV counsellors—were not present. Similar results were found in a study in Zambia, where providers trained in couples' voluntary HTC increased their knowledge scores by 22% [24]. Furthermore, the fact that yield increased significantly among women, but not men, suggests that the training may be particularly effective in aiding diagnosis among a population with a low level of undiagnosed HIV. Conversely, men had greater HIV positivity yield values than women in both pre- and post-training phases, which is consistent with findings from a study performed in Mozambique in 2019 [25] and could be explained by the higher prevalence of undiagnosed HIV among men (11.4% vs 9.4% in men and women respectively). Men are less likely to be tested and they have delayed entry to HIV care (first 95) in SSA countries like Mozambique [15, 26], leading to a higher number of undiagnosed men living with HIV.

Yield serves as an operational indicator of the performance of targeted PITC; however, an increase in yield does not always correlate with a rise in the absolute number of new HIV diagnosis. In this study, despite observing a higher yield after the training, the absolute count of HIV-positive individuals decreased post-training (146 compared to 165 pre-training). Notably, while 3000 more individuals sought healthcare during the post-training observation phase, 1000 fewer were referred and tested compared to the pre-training phase. This highlights how providers were more successful in identifying HIV-positive individuals after receiving the targeted PITC training module in a limited-resource rural setting where universal PITC is not feasible. This becomes crucial in striving toward achieving the UNAIDS 95-95-95 targets, emphasizing the need for continuous identification of people living with HIV, especially amid diminishing resources.

The rise in HIV positivity yield post-training was not observed in the ED, a health department with a low volume of clients but a 20% prevalence of undiagnosed HIV in this study.

However, presenting at ED was associated with a positive HIV test. WHO guidelines recommend universal PITC in emergency services, and its implementation and enforcement could prevent missed opportunities for HIV diagnosis in Mozambique [27, 28]. Studies in other countries, such as Tanzania, have shown that universal HIV testing in the ED is feasible with high acceptance and yield [29, 30].

Furthermore, the study found that most risk factors, signs and symptoms associated with a positive HIV test were previously described in the literature [9, 31, 32] and included in the Mozambican targeted PITC algorithm, which identified 72% of the positive HIV tests. Other variables such as age, occupation, unawareness of partner's HIV status and recent visits to a healer were also found to be associated with a positive HIV test, aligning with prior research [9, 31, 33, 34]. Considering the potential impact of these factors on the performance of targeted PITC, further investigations are recommended to evaluate the inclusion of these factors into the existing targeted PITC algorithm. Moreover, the current targeted PITC algorithm excludes individuals with a negative HIV test within three months prior. This exclusion criterion takes precedence over the inclusion criteria of HIV risk factors and symptoms. However, our findings show that among these algorithm non-eligible individuals, 3.5% who tested positive had at least one HIV risk factor or symptom included in the algorithm. These cases reflect new infections within the three-month window, emphasizing the need for further research to evaluate the potential removal of this exclusion criterion from the algorithm, especially in areas with high HIV incidence.

Very few studies have evaluated the effect of a targeted PITC training on HIV positivity yield. To our knowledge, this is the first evaluation of a MoH training module performed in Mozambique. One major strength was the inclusion of pre- and post-training routine data analysis to estimate yield in real-world conditions. In contrast, there were several limitations due to the design of the study. The lack of a control group restricted the ability to draw causal inferences or account for secular trends [35]. Although the time interval between pre- and post- training was short, we were not able to infer whether the increase in yield seen in the post-training routine phase was primarily caused by the training module, or also by others factors that might be temporally coincident with it. Similarly, we could not control for fluctuations in the number of healthcare seekers or positive HIV tests observed in all phases, which could be due to seasonal patterns in clinical activities and health-seeking behaviour previously described in similar settings [36–38]. Additionally, due to the nature of health system aggregated data collection for the routine phases, paired data analysis was not possible, which may have hindered the control of differences between healthcare providers. More ideal study designs, such as cluster randomised trials or stepped-wedge at healthcare facility level in order to incorporate control facilities, were not feasible since the MoH training was about to be implemented and resources were limited. Lastly, while our study included only medical assistants as healthcare providers, it is worth noting that extending the training to individuals from diverse professional backgrounds may lead to varied impacts on HIV positivity yield, influenced by their prior experience.

## Conclusions

Targeted PITC's performance, as measured by HIV positivity yield, improved among women in a high-volume urgent care setting (triage) in the months following a MoH targeted PITC training study in a resource-constrained rural area burdened by high HIV prevalence. Regular refresher trainings could enhance targeted PITC yield sustainably over time, and thus, further studies are needed to determine how to introduce the training, in terms of length and frequency. Additionally, our findings underscore the necessity of re-evaluating the current

targeted PITC algorithm by considering to remove the recent HIV-negative test criterion for those with HIV risk factors and add factors such as age, occupation, unawareness of partner's HIV status, and visits to a traditional healer. Adapting existing algorithms is crucial for reaching undiagnosed PLHIV, treating all PLHIV and eliminating HIV.

## Supporting information

**S1 Fig. Targeted PITC algorithm implemented by the Ministry of Health (MoH) in Mozambique.** Figure adapted from the Differentiated Services Delivery Models Guidelines by the Mozambican MoH, 2018. Abbreviations: ART: antiretroviral therapy, PITC: provider-initiated testing and counselling.
(TIF)

**S1 Table. Factors associated with a positive HIV test.** Adjusted odds ratios from logistic regression analyses by sex. [a] Client participants tested for HIV in both observation phases. 14 individuals with an undetermined HIV test result were excluded. For the multivariable analysis, 1471 men and 2422 were included, respectively. [b] Sociodemographic characteristics, risk factors or signs and symptoms not included in the national targeted PITC algorithm of Mozambique. [c] Occasionally: less than half of the times; Frequently: more than half of the times [d] The aOR among women could not be estimated because there was not any woman included in the multivariable analysis who presented with night sweats. Abbreviations: aOR: adjusted odds ratio, CI: confidence interval, PITC: provider-initiated testing and counselling, Ref.: reference category.
(DOCX)

## Acknowledgments

The authors gratefully acknowledge the staff at Centro de Investigação em Saúde de Manhiça, in the Manhiça District, Mozambique, who worked to collect and manage the data, the Ministry of Health of Mozambique, our research team, collaborators, and especially all communities and participants involved.

## Author Contributions

**Conceptualization:** Sheila Fernández-Luis, Tacilta Nhampossa, Laura Fuente-Soro, Elisa López-Varela, Edson Bernardo, Stanley C. Wei, Peter Kerndt, Peter Young, Denise Naniche.

**Data curation:** Anna Saura-Lázaro, Sheila Fernández-Luis, Laura Fuente-Soro, Orvalho Augusto, Teresa Sánchez.

**Formal analysis:** Anna Saura-Lázaro, Sheila Fernández-Luis, Orvalho Augusto, Teresa Sánchez.

**Funding acquisition:** Elisa López-Varela, Denise Naniche.

**Methodology:** Anna Saura-Lázaro, Sheila Fernández-Luis, Tacilta Nhampossa, Laura Fuente-Soro, Elisa López-Varela, Edson Bernardo, Orvalho Augusto, Stanley C. Wei, Peter Kerndt, Denise Naniche.

**Project administration:** Sheila Fernández-Luis, Laura Fuente-Soro.

**Resources:** Fernando Boene.

**Supervision:** Tacilta Nhampossa, Elisa López-Varela, Denise Naniche.

**Writing – original draft:** Anna Saura-Lázaro, Sheila Fernández-Luis, Teresa Sánchez.

**Writing – review & editing:** Anna Saura-Lázaro, Sheila Fernández-Luis, Tacilta Nhampossa, Laura Fuente-Soro, Elisa López-Varela, Edson Bernardo, Orvalho Augusto, Paula Vaz, Stanley C. Wei, Peter Kerndt, Nely Honwana, Peter Young, Guita Amane, Fernando Boene, Denise Naniche.

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
