## [Decision Letter · Decision Letter 0]

29 Nov 2023

PONE-D-23-28399Enhancing HIV testing yield in southern Mozambique: the effect of a Ministry of Health training module in targeted provider-initiated testing and counsellingPLOS ONE

Dear Dr. Saura-Lázaro,

Thank you for submitting your manuscript to PLOS ONE. After careful consideration, we feel that it has merit but does not fully meet PLOS ONE’s publication criteria as it currently stands. Therefore, we invite you to submit a revised version of the manuscript that addresses the points raised during the review process.

We look forward to receiving your revised manuscript.

Kind regards,

Hamufare Dumisani Dumisani Mugauri, Ph.D. Public Health

Academic Editor

PLOS ONE

“We acknowledge support from the grant CEX2018-000806-S funded by MCIN/AEI/ 10.13039/501100011033, and support from the Generalitat de Catalunya through the CERCA Programme. The authors gratefully acknowledge the staff at Centro de Investigação em Saúde de Manhiça, in the Manhiça District, Mozambique, who worked to collect and manage the data, the Ministry of Health of Mozambique, our research team, collaborators, and especially all communities and participants involved.”

“This publication has been supported by the President’s Emergency Plan for AIDS Relief (PEPFAR) through the Centers for Disease Control and Prevention (CDC) under the terms of NU2GGH002092, the Severo Ochoa predoctoral fellowship by the Barcelona Institute of Global Health (ISGlobal) to ASL, the predoctoral fellowship from the Secretariat of Universities and Research, Ministry of Enterprise and Knowledge of the Government of Catalonia and cofounded by European Social Fund to ASL and SFL, and the European Respiratory Society (ERS) and the European Union (EU)’s H2020 research and innovation programme under the Marie Sklodowska-Curie grant agreement [847462] to ELV (This publication reflects only the author's view. The ERS, Research Executive Agency and EU are not responsible for any use that may be made of the information it contains). SW, employed by CDC participated in the conceptualization, study design and manuscript revision. For the remaining authors none were declared.

The findings and conclusions in this report are those of the author(s) and do not necessarily represent the official position of the funding agencies.”

Reviewers' comments:

Reviewer's Responses to Questions

**Comments to the Author**

1. Is the manuscript technically sound, and do the data support the conclusions?

Reviewer #1: Yes

Reviewer #2: Partly

2. Has the statistical analysis been performed appropriately and rigorously? 

Reviewer #1: Yes

Reviewer #2: Yes

3. Have the authors made all data underlying the findings in their manuscript fully available?

Reviewer #1: Yes

Reviewer #2: Yes

4. Is the manuscript presented in an intelligible fashion and written in standard English?

Reviewer #1: Yes

Reviewer #2: Yes

5. Review Comments to the Author

Reviewer #1: The study brings a very important message in reaching the last mile of the first 95% UNAIDS target, particulary for Mozambique where the HIV burden is still high. However, as a general comment I think the generalizability of the study results is compromised first because the study was limited to a single geographical area in Mozambique (Mozambique is a sizeable country with a lot of diversity, even within the health sector). Second, the study did not include a control group which would have allowed for a more meaningful comparison. Finally, I think if the study had been conducted over a long period of time, one could have been able to assess long term trends and more accurate predictions.

However, I applaud the investigators as despite the limitations the study raises some important points for programs to consider: 1) if the current algorithms are still adequate for the last mile, 2) the current training is it enough, frequency, quality and 3) application of evidence based strategies. For this I believe this article is very important.

Some few comments:

- Was the training online or was it in person, what was the duration of the training and how many health professionals were trained?

- The study was conducted in a rural area in Maputo province. Could the authors explain the choice of the health facilities? Why were some urban health facilities in Maputo not included?

- Could the authors please specify the professional level of the health professionals in the triage and emergency department areas? Could this have influenced in the results for example if some facilities had doctors vs nurses in triage?

- Line 417 - included references twice, please delete one

Reviewer #2: Global overview

This is an interesting study measuring the effectiveness of a training module to improve the yield of PITC in triage and emergency departments using a pre-post study design. While acknowledging the significance of the study, I find that the manuscript lacks focus and requires a more cautious interpretation of some results.

Since their main objective is to enhance “positivity yield” (i.e., the sentitivity of the screening algorithm), I do believe they should consider the volume of new positive cases in the assessment. I understand that we want to improve the yield to reduce time burden and cost of doing systematic test but this should not be at the expense of the volume of newly diagnosed individuals (e.g., for a same denominator, it is better to have a yield of 5% if it leads to 100 new cases of HIV positivity rather than a yield of 10% and only 10 new cases of HIV positivity). Usually, this information would be given by sensibility (which would have provided the number of true HIV negative excluded), but the current study design does not allow to infer on that aspect. Thus, I believe this is important to provide alongside those yields, the number of new cases with the number of eligible population (e.g., patient seen in emergency services).

The authors have included a secondary objective which aims to improve the current PITC algorithm. I am not comfortable with that analysis as I have some doubts about the generalisation of their results. There is not much description for the methods of that secondary objective especially on how the 509 positives tests have been included in the analysis (during the intervention, before or both?), and the fact that this population has been already selected through provider perceptions or PITC algorithm can potentially create a selection bias. I would suggest removing that part to refocus on the main objective of the paper or add sufficient information to be able to assess those results and be more cautious on the interpretation of the results.

Title

The title should accurately reflect the paper's focus on "positivity yield" rather than "HIV testing yield". This terminology should be corrected throughout their manuscript

Abstract

Since the paper is about the positivity yield in four specific periods (pre-post intervention during routine practice and pre-post intervention during the study period), I think the authors should focus more on that specific aspect and present yields with numerators and denominators for each of the 4 periods.

Background

While the background is informative, it lacks references on the effectiveness of targeted PITC. It misses some key references about the effectiveness of targeted PITC (see one suggested below among many other ones).

Leblanc J, Hejblum G, Costagliola D, Durand-Zaleski I, Lert F, de Truchis P, Verbeke G, Rousseau A, Piquet H, Simon F, Pateron D, Simon T, Crémieux AC; DICI-VIH (Dépistage Infirmier CIblé du VIH) Group. Targeted HIV Screening in Eight Emergency Departments: The DICI-VIH Cluster-Randomized Two-Period Crossover Trial. Ann Emerg Med. 2018 Jul;72(1):41-53.e9. doi: 10.1016/j.annemergmed.2017.09.011. Epub 2017 Oct 31. PMID: 29092761.

Methods

A description of the control period, especially how PITC was implemented before the intervention, would enhance the understanding of the study design.

Results

Figure 2 resolution is very low, I was not able to read it.

It would have been interesting to have a figure should the evolution of those positive yield per month or every two months to see the eventual fluctuation of positivity yield.

I am not certain to understand the drop of patients seen during the study period (n=7,102, ~1776 patients seen per month) compared to the routine period (n=33,261, ~ 4,158 per month).

Discussion

The discussion is mainly focusing on improving the PITC algorithm which is not the main topic on the paper.

Since the intervention had no effect during the study period but was found effective between the pre and post routine period, I think a broader discussion about that aspect (e.g., why this intervention did not work in the intervention period? Were there any study contaminations?) should be more highlighted here with the potential implication of such results.

6. PLOS authors have the option to publish the peer review history of their article (what does this mean?). If published, this will include your full peer review and any attached files.

Reviewer #1: No

Reviewer #2: No

---

## [Author Response · Author response to Decision Letter 0]

17 Jan 2024

Re: Response to the editorial team and reviewers - PONE-D-23-28399

Dear editorial team and reviewers, 

We would like to express our gratitude to the editorial team and reviewers for their thorough review of the manuscript. We have carefully read through and considered all comments and have found them very useful. We have revised the manuscript incorporating the revisions and have provided clean and tracked file versions of the revised manuscript, along with this detailed response to the comments. 

Response to editor

RE: We appreciate the feedback regarding the adherence to PLOS ONE style requirements. After carefully reviewing the provided PLOS ONE style templates for manuscript formatting, we have meticulously aligned our manuscript with these guidelines. Our revisions involve adding level 1 and level 2 headings, applying indentation in each paragraph, including supplementary file legends in the Supporting information section of the main manuscript, and comprehensive revising file naming. 

RE: Thank you for the feedback on the Ethics statement and participant consent details. We have revised and updated the Ethics statement in the Methods section (line 230), addressing the approach for both observation and routine study phases: "All study participants, both healthcare providers and individuals presenting at the healthcare facilities during the observation phases of the study, completed written IC. For participants aged 15-18 years, a parent/legal representative's additional consent was necessary. Regarding the routine phases, as we extracted de-identified aggregated data, informed consent was not required". Should there be a necessity for further clarification or additional information on participant consent, we are prepared to provide any necessary details as guided by your instructions.

3. Thank you for stating the following in the Acknowledgments Section of your manuscript: “We acknowledge support from the grant CEX2018-000806-S funded by MCIN/AEI/ 10.13039/501100011033, and support from the Generalitat de Catalunya through the CERCA Programme. The authors gratefully acknowledge the staff at Centro de Investigação em Saúde de Manhiça, in the Manhiça District, Mozambique, who worked to collect and manage the data, the Ministry of Health of Mozambique, our research team, collaborators, and especially all communities and participants involved.”

Please remove any funding-related text from the manuscript and let us know how you would like to update your Funding Statement. Currently, your Funding Statement reads as follows: “This publication has been supported by the President’s Emergency Plan for AIDS Relief (PEPFAR) through the Centers for Disease Control and Prevention (CDC) under the terms of NU2GGH002092, the Severo Ochoa predoctoral fellowship by the Barcelona Institute of Global Health (ISGlobal) to ASL, the predoctoral fellowship from the Secretariat of Universities and Research, Ministry of Enterprise and Knowledge of the Government of Catalonia and cofounded by European Social Fund to ASL and SFL, and the European Respiratory Society (ERS) and the European Union (EU)’s H2020 research and innovation programme under the Marie Sklodowska-Curie grant agreement [847462] to ELV (This publication reflects only the author's view. The ERS, Research Executive Agency and EU are not responsible for any use that may be made of the information it contains). SW, employed by CDC participated in the conceptualization, study design and manuscript revision. For the remaining authors none were declared.

The findings and conclusions in this report are those of the author(s) and do not necessarily represent the official position of the funding agencies.”

RE: We appreciate the guidance on the appropriate placement of funding information in our manuscript. The funding acknowledgement mentioned in this sentence “We acknowledge support from the grant CEX2018-000806-S funded by MCIN/AEI/ 10.13039/501100011033, and support from the Generalitat de Catalunya through the CERCA Programme”, is not directly associated with the funding for this study. Our institution requires us to include this information in the acknowledgements. However, as requested by the editor, we have removed it from the manuscript and reallocated it at the end of the funding statement.

Therefore, the final funding statement will read as follows: “This publication has been supported by the President’s Emergency Plan for AIDS Relief (PEPFAR) through the Centers for Disease Control and Prevention (CDC) under the terms of NU2GGH002092, the Severo Ochoa predoctoral fellowship by the Barcelona Institute of Global Health (ISGlobal) to ASL, the predoctoral fellowship from the Secretariat of Universities and Research, Ministry of Enterprise and Knowledge of the Government of Catalonia and cofounded by European Social Fund to ASL and SFL, and the European Respiratory Society (ERS) and the European Union (EU)’s H2020 research and innovation programme under the Marie Sklodowska-Curie grant agreement [847462] to ELV (This publication reflects only the author's view. The ERS, Research Executive Agency and EU are not responsible for any use that may be made of the information it contains). SW, employed by CDC participated in the conceptualization, study design and manuscript revision. For the remaining authors none were declared. The findings and conclusions in this report are those of the author(s) and do not necessarily represent the official position of the funding agencies. Lastly, we acknowledge support from the grant CEX2018-000806-S funded by MCIN/AEI/ 10.13039/501100011033, and support from the Generalitat de Catalunya through the CERCA Programme.”

RE: Thank you for bringing up the concern regarding the data availability statement. In response to your guidance, we propose the following updates: “The data cannot be publicly shared due to ethical restrictions. Data contain potentially sensitive information, and the national ethics committee (CNBS) does not authorize data sharing without a protocol request specifying the objectives and the researchers who will have access to the data. The datasets generated and/or analysed during the current study are available upon request (contact via llorenc.quinto@isglobal) for researchers who meet the criteria for accessing confidential data.”

Response to reviewers

Reviewer #1:

Review Comments to the Author

The study brings a very important message in reaching the last mile of the first 95% UNAIDS target, particularly for Mozambique where the HIV burden is still high. However, as a general comment I think the generalizability of the study results is compromised first because the study was limited to a single geographical area in Mozambique (Mozambique is a sizeable country with a lot of diversity, even within the health sector). Second, the study did not include a control group which would have allowed for a more meaningful comparison. Finally, I think if the study had been conducted over a long period of time, one could have been able to assess long term trends and more accurate predictions.

However, I applaud the investigators as despite the limitations the study raises some important points for programs to consider: 1) if the current algorithms are still adequate for the last mile, 2) the current training is it enough, frequency, quality and 3) application of evidence based strategies. For this I believe this article is very important.

RE: Thank you for the comprehensive overview and recognizing the study's importance in shaping HIV programs toward achieving the last mile of the first 95 UNAIDS target. We acknowledge that the absence of a control group may impact the ability of draw causal inferences. This limitation has been appropriately addressed and discussed in the manuscript's Discussion section at line 482. Regarding the duration of the study, as our analysis did not focus on trends, this factor did not impact our findings. However, as this is a first step, we recognize the importance of conducting a longer-term study to analyse future trends. 

Concerning the study's restriction to a rural setting, we acknowledge the reviewer's point about potential limitations in generalizing the results to urban areas. However, as elaborated further in a subsequent comment, our intentional focus was on a rural area marked by a significant HIV burden and constrained resources

Some few comments:

- Was the training online or was it in person, what was the duration of the training and how many health professionals were trained?

RE: Thank you for the comment. The training sessions covering signs, symptoms, and risk factors for HIV (targeted PITC) were conducted in person by the HIV testing and counselling reference group from the Mozambican Ministry of Health at the Manhiça District hospital. These details have been added in line 183 of the manuscript. The training spanned one week, from the 13th to the 19th of May (refer to Figure 1 for clarification). The 19 healthcare professionals who participated in the study were the same individuals who attended this training module, as mentioned in lines 149 and 235, having agreed to take part in both study phases and the training.

- The study was conducted in a rural area in Maputo province. Could the authors explain the choice of the health facilities? Why were some urban health facilities in Maputo not included?

RE: Thank you for the insightful comment. The authors chose to focus the study on a semi-rural area due to its high HIV prevalence and the strain on its healthcare system, which operates with limited resources. These details have been included in the Methods section at line 134. Our aim was to contribute insights into rural areas, often overshadowed by urban areas, despite their crucial role in the “last mile” efforts targeting harder-to-reach populations. We have made specific modifications throughout the manuscript (lines 51, 71, 123, 385, 431 and 501) to emphasize that our study setting and findings are centred on rural areas. Additionally, our team has established a longstanding collaboration with the local HIV program implementer, Fundação Ariel, which significantly supported our research efforts in this area. The selection of the healthcare facilities within the study area was based on patient volume, including the four highest-volume facilities in the district (line 132). 

Acknowledging Mozambique's diverse healthcare landscape, as highlighted by the reviewer in the overview, we recognize that our specific site selection might limit generalizability to urban areas. However, given our study’s focus on rural challenges, the authors do not consider this a limitation and believe our findings hold potential applicability to other rural areas facing similar challenges of a high HIV burden and constrained resources.

- Could the authors please specify the professional level of the health professionals in the triage and emergency department areas? Could this have influenced in the results for example if some facilities had doctors vs nurses in triage?

RE: We appreciate the comment. All 19 healthcare providers included in the study were medical assistants (line 235), who, in Mozambique, undergo four years training in basic medical care, allowing them to prescribe medicines and support minor surgeries. This uniform professional level minimised the potential influence of professional level diversity in our analysis. However, as the reviewer infers, if the training were applied to healthcare workers with variations in professional levels, the impact of training on HIV testing yield could differ based on the pre-training experience and background of individuals. We have included this important observation in the Discussion section at line 493. It is important to note that HIV testing referral tasks in Mozambique are primarily carried out by medical assistants and nurses. 

- Line 417 - included references twice, please delete one

RE: Thank you for highlighting the error. We have now removed the duplicated references as per your observation.

Reviewer #2:

Review Comments to the Author

Global overview

This is an interesting study measuring the effectiveness of a training module to improve the yield of PITC in triage and emergency departments using a pre-post study design. While acknowledging the significance of the study, I find that the manuscript lacks focus and requires a more cautious interpretation of some results.

Since their main objective is to enhance “positivity yield” (i.e., the sensitivity of the screening algorithm), I do believe they should consider the volume of new positive cases in the assessment. I understand that we want to improve the yield to reduce time burden and cost of doing systematic test but this should not be at the expense of the volume of newly diagnosed individuals (e.g., for a same denominator, it is better to have a yield of 5% if it leads to 100 new cases of HIV positivity rather than a yield of 10% and only 10 new cases of HIV positivity). U

---

## [Decision Letter · Decision Letter 1]

20 Mar 2024

PONE-D-23-28399R1Enhancing HIV positivity yield in southern Mozambique: the effect of a Ministry of Health training module in targeted provider-initiated testing and counsellingPLOS ONE

Dear Dr. Saura-Lázaro,

Thank you for submitting your manuscript to PLOS ONE. After careful consideration, we feel that it has merit but does not fully meet PLOS ONE’s publication criteria as it currently stands. Therefore, we invite you to submit a revised version of the manuscript that addresses the points raised during the review process.

We look forward to receiving your revised manuscript.

Kind regards,

Hamufare Dumisani Dumisani Mugauri, Ph.D. Public Health

Academic Editor

PLOS ONE

Reviewers' comments:

Reviewer's Responses to Questions

**Comments to the Author**

1. If the authors have adequately addressed your comments raised in a previous round of review and you feel that this manuscript is now acceptable for publication, you may indicate that here to bypass the “Comments to the Author” section, enter your conflict of interest statement in the “Confidential to Editor” section, and submit your "Accept" recommendation.

Reviewer #1: All comments have been addressed

Reviewer #3: (No Response)

2. Is the manuscript technically sound, and do the data support the conclusions?

Reviewer #1: Yes

Reviewer #3: Partly

3. Has the statistical analysis been performed appropriately and rigorously? 

Reviewer #1: Yes

Reviewer #3: Yes

4. Have the authors made all data underlying the findings in their manuscript fully available?

Reviewer #1: Yes

Reviewer #3: No

5. Is the manuscript presented in an intelligible fashion and written in standard English?

Reviewer #1: Yes

Reviewer #3: Yes

6. Review Comments to the Author

Reviewer #1: All questions were adequately addressed and the authors made significant improvement to the manuscript. It is much better and clearer.

Reviewer #3: This is an interesting paper on the impact of training of health care providers in a primary care seting in rural Mozambique on targeted provider initiated HIV testing and counselling (PITC) and assessing the impact on HIV positivity yield among those in care within the study period (e.g. consented participants in care before and after the study period) and in the wider timeframe of 4 months prior to and after completion of the study looking at aggregate routine care data.

This submission had already responded to some reviewers comments.

Overall the study design and conclusions are sound although the way the results are presented and summarised is rather confusing and difficult to follow and would suggest some edits to address this.

Major comments:

Abstract

• It would be helpful for the reader to clarify the primary versus secondary outcomes of interest in this study and signpost these results accordingly. For example, I assume the primary outcomes are those based on the study ‘observation phases’ (n=7102, main outcome is HIV positivity yield overall and by sex). Secondary objectives – focusing on outcomes in the ‘routine data phases’? If so, in the results the authors should provide the N for both primary and secondary objectives, in the pre and post period, with the sex distribution and proportion recruited at triage for consistency, before reporting the yield. It would also be important to include data on the number/proportion in care targeted for HIV testing before/after the study period to show decline in numbers tested in the post training period.

• Also important to state clearly that the analyses on factors associated with positive HIV test presented here is from the study period and not the routine data phase as it follows immediately after reporting the yield of the routine data phase.

Methods and results

• In the methods it states at during the study period, all eligible patients in care were asked to consent to be part of this study prior to being assessed for PITC and all offered HIV counselling and testing, irrespective of PITC recommendations to assess how many new HIV diagnoses were missed by targeted PITC. However, the authors do not describe how these data are used and compared to the number of new diagnoses captured through targeted PITC in the statistical analysis section of the methods. This seems to be a key strength in the study design but does not really stand out as a key result?

• Results line 256-258: you provide the proportion of study participants in the study observation period who meet the PITC algorithm criteria and the % not meeting the criteria but still referred for PITC. It would be interesting to see if many of those not meeting the criteria had characteristics consistent with those identified in the regression analysis as being associated with HIV positive result?

• Table 1 summarises the characteristics of all participants enrolled in the study n=7102, although only a subgroup were selected for PITC, and the characteristics of those selected are not described. It would be more informative to expand on this table 1A, and give also the characteristics of the following subgroups within each pre/post observation phase groups (i) those selected for PITC and met the criteria of the algorithm, (ii) those selected for PITC but not met criteria of algorithm, (iii) and those not referred for PITC and for all subgroups include the number testing HIV positive in the table.

• Results line 296-300 and Table 2: this summarises the total referred for PITC in the pre/post observation period and number tested positive. What is missing is the number of study participants not referred to PITC who were tested positive e.g. missed new diagnoses. Where are these data? Would be particularly interesting to see by department, especially emergency where yield appears highest? Also it should be clearly stated in the results and discussion that these yield ratios are unadjusted for potentially varying characteristics of clients over time.

• Results: page 36 first 2 paragraphs discusses the increase in yield in the triage department but not the emergency department. It is worth noting the very high yield in the observation period overall at >15% and trends suggesting a reduction in yield post training although not statistically significant, likely due to smaller samples. Here in particular it would be interesting to know how many diagnoses were missed by PITC?

• Factors associated with positive HIV test: this analysis appears to be using data from all clients enrolled in the study and not just those targeted for PITC. This is where it gets rather confusing for the reader as the paper on one hands presents yield of targeted PITC with a subgroup selected for PITC, and then factors associated with positive result within a bigger subgroup of n=5028 whose characteristics are not presented in any of the previous tables.

• Results, page 45, line 361: the coauthors state targeted PITC would have identified 72% of all persons with a positive result (306 of 509), but it is not clear if most of these were picked up by the healthcare workers who referred them for PITC despite not meeting the algorithm criteria?

Figure 2 is very unclear - impossible to read.

7. PLOS authors have the option to publish the peer review history of their article (what does this mean?). If published, this will include your full peer review and any attached files.

Reviewer #1: No

Reviewer #3: No

---

## [Author Response · Author response to Decision Letter 1]

17 Apr 2024

Dear editorial team and reviewers, 

We would like to express our gratitude to the editorial team and reviewers for their thorough review of the first round of manuscript revisions. We have carefully read through and considered all comments and have found them very useful. We have revised the manuscript incorporating the revisions and have provided clean and tracked file versions of the revised manuscript, along with this detailed response to the comments. Additionally, we would like to inform the editor that we have expanded the abstract by 39 word to incorporate the information requested by the reviewer. However, we remain open to reducing this new information, should the editor deem it necessary. 

Response to reviewers

Reviewer #1:

All questions were adequately addressed and the authors made significant improvement to the manuscript. It is much better and clearer.

Reviewer #3

This is an interesting paper on the impact of training of health care providers in a primary care seting in rural Mozambique on targeted provider initiated HIV testing and counselling (PITC) and assessing the impact on HIV positivity yield among those in care within the study period (e.g. consented participants in care before and after the study period) and in the wider timeframe of 4 months prior to and after completion of the study looking at aggregate routine care data. 

This submission had already responded to some reviewers comments. 

Overall the study design and conclusions are sound although the way the results are presented and summarised is rather confusing and difficult to follow and would suggest some edits to address this. 

Major comments: 

Abstract

• It would be helpful for the reader to clarify the primary versus secondary outcomes of interest in this study and signpost these results accordingly. For example, I assume the primary outcomes are those based on the study ‘observation phases’ (n=7102, main outcome is HIV positivity yield overall and by sex). Secondary objectives – focusing on outcomes in the ‘routine data phases’? If so, in the results the authors should provide the N for both primary and secondary objectives, in the pre and post period, with the sex distribution and proportion recruited at triage for consistency, before reporting the yield. It would also be important to include data on the number/proportion in care targeted for HIV testing before/after the study period to show decline in numbers tested in the post training period. 

RE: Thank you for the comment. The primary objective of our study was to evaluate the effect of a new Ministry of Health training module in targeted PITC on HIV positivity yield, which we identified as our primary outcome. For this evaluation we used data from both the observation and routine phases, with the latter phase serving to control for potential observer bias (Methods section, line 151). We have clarified the primary objective and outcome in the Introduction section at line 121 and in the Methods section at line 206. Our secondary objective was to identify factors associated with a positive HIV test, with odds ratios from logistic regression models serving as the secondary outcome. The secondary objective is detailed in the Introduction section at line 123 and the secondary outcome has been further clarified in the Methods section at line 214. In the abstract, we have incorporated the statistical analysis used for the secondary objective at line 56 to enhance clarity. Additionally, acknowledging the importance of consistency, and following the reviewer’s suggestion, we have now included in the abstract, at line 59, the number of participants (N), broken down by sex and department, for the routine phases. Lastly, the number of individuals in care referred for HIV testing, as suggested in the first round of revisions, was included as the denominator for the HIV positivity yield results.

• Also important to state clearly that the analyses on factors associated with positive HIV test presented here is from the study period and not the routine data phase as it follows immediately after reporting the yield of the routine data phase. 

RE: We appreciate the comment and agree with the reviewer’s suggestion. Accordingly, we have clarified in the Abstract at line 56 that the analysis of factors associated with a positive HIV test was based on data from the observation phases. 

Methods and results 

• In the methods it states at during the study period, all eligible patients in care were asked to consent to be part of this study prior to being assessed for PITC and all offered HIV counselling and testing, irrespective of PITC recommendations to assess how many new HIV diagnoses were missed by targeted PITC. However, the authors do not describe how these data are used and compared to the number of new diagnoses captured through targeted PITC in the statistical analysis section of the methods. This seems to be a key strength in the study design but does not really stand out as a key result?

RE: Thank you for the feedback and our apologies for any confusion caused. The data highlighted by the reviewer are indeed crucial and presented in Figure 2. Following the suggestion in the first round of revisions, we improved the resolution of Figure 2, which contains essential information. However, it appears that it was not legible in the PDF generated by the submission system and may need to be downloaded directly from the system. To aid in the review, we have attached Figure 2 to this letter. These findings are also presented in the Results section at line 366: “Among all participants who tested positive for HIV, 72.0% (364 out of 509) were eligible for testing under the current targeted PITC algorithm. This leaves 145 HIV diagnoses that were missed by the existing algorithm”. Additionally, per reviewer’s advice, we have further detailed how these data were analysed in the statistical analysis in the Methods section at line 217. 

• Results line 256-258: you provide the proportion of study participants in the study observation period who meet the PITC algorithm criteria and the % not meeting the criteria but still referred for PITC. It would be interesting to see if many of those not meeting the criteria had characteristics consistent with those identified in the regression analysis as being associated with HIV positive result? 

RE: Thank you for the comment. In our analysis addressing the secondary objective—to identify factors associated with a positive HIV test—we included all tested participants, irrespective of whether they were referred for HIV testing by the healthcare provider through targeted PITC (Methods section, line 192). This approach allowed us to uncover additional factors that are not considered in the current targeted PITC algorithm but could potentially increase the detection of HIV diagnoses. Indeed, we identified several factors associated with a positive HIV test that, if included as eligibility criteria, could help identify additional HIV-positive individuals deemed non-eligible by the current targeted PITC algorithm. These exploratory findings are detailed in the Results section at line 373, with further clarification provided at lines 366 and 375.

• Table 1 summarises the characteristics of all participants enrolled in the study n=7102, although only a subgroup were selected for PITC, and the characteristics of those selected are not described. It would be more informative to expand on this table 1A, and give also the characteristics of the following subgroups within each pre/post observation phase groups (i) those selected for PITC and met the criteria of the algorithm, (ii) those selected for PITC but not met criteria of algorithm, (iii) and those not referred for PITC and for all subgroups include the number testing HIV positive in the table.

RE: Thank you for the suggestion. As previously mentioned, the primary objective of our study was to evaluate the HIV positivity yield before and after implementing a training module in targeted PITC. Accordingly, we stratified the data in Table 1 to compare clients’ characteristics pre- and post-training phases, to ensure the comparability of groups. We consider that adding further stratification by additional subgroups might make the table’s interpretation more challenging and not shed further light on our primary or secondary objectives. However, we acknowledge the reviewer’s point regarding the importance of presenting the actual figures within each subgroup. And indeed, this is presented in Figure 2, which stratifies participants by their eligibility for HIV testing through targeted PITC and by healthcare providers referrals. 

• Results line 296-300 and Table 2: this summarises the total referred for PITC in the pre/post observation period and number tested positive. What is missing is the number of study participants not referred to PITC who were tested positive e.g. missed new diagnoses. Where are these data? Would be particularly interesting to see by department, especially emergency where yield appears highest? Also it should be clearly stated in the results and discussion that these yield ratios are unadjusted for potentially varying characteristics of clients over time.

RE: Thank you for the comment. The data regarding study participants who were not referred for HIV testing through targeted PITC by healthcare providers but tested positive are shown in Figure 2, enclosed above for reference. Regarding stratification by department, it is important to note that all participants enrolled in the emergency department who tested positive during the observation phases, including both pre- and post-training phases (n=49, inferred from data from Tables 2B and 3), were referred for HIV testing through targeted PITC by healthcare providers. This referral pattern was significantly influenced by observer bias, the Hawthorne effect—a point thoroughly discussed in the manuscript. Additionally, the higher HIV prevalence and generally more severe conditions of patients attending the emergency department likely contributed to the high frequency of referrals for HIV testing by healthcare providers. This context helps explain the high HIV positivity yield observed in this department and why there was no increase in HIV positivity yield after the training module, given that all positive individuals were referred by the provider during both observation phases. Consequently, the specific question raised by the reviewer regarding missed diagnosis by the healthcare providers through targeted PITC in the emergency department cannot be addressed. Additionally, the issue of unadjusted yield ratios has been recognized and is discussed as a limitation in the Discussion section at line 459.

• Results: page 36 first 2 paragraphs discusses the increase in yield in the triage department but not the emergency department. It is worth noting the very high yield in the observation period overall at >15% and trends suggesting a reduction in yield post training although not statistically significant, likely due to smaller samples. Here in particular it would be interesting to know how many diagnoses were missed by PITC?

RE: Thank you for the suggestion. As detailed in our previous response, all participants who tested positive for HIV in the emergency department during both pre- and post-training observation phases were referred for HIV testing by healthcare providers. This referral pattern was primarily influenced by observer bias among healthcare providers and the generally higher severity of conditions in patients attending this department. Therefore, the specific question raised by the reviewer regarding missed diagnosis by the healthcare providers through targeted PITC in the emergency department cannot be addressed. 

• Factors associated with positive HIV test: this analysis appears to be using data from all clients enrolled in the study and not just those targeted for PITC. This is where it gets rather confusing for the reader as the paper on one hands presents yield of targeted PITC with a subgroup selected for PITC, and then factors associated with positive result within a bigger subgroup of n=5028 whose characteristics are not presented in any of the previous tables. 

RE: Thank you for the comment, and we apologise for any confusion caused. During the observation phases, we offered HIV testing to all individuals presenting at the healthcare facility who accepted to participate in the study, driven by two key reasons: i) to identify HIV diagnoses missed by the current targeted PITC algorithm, and ii) to explore factors associated with a positive HIV test beyond those included in the current targeted PITC algorithm, necessitating the inclusion of not just patients referred for targeted PITC to minimise potential selection bias. We have detailed this approach in the Methods section at line 192. Additionally, it is important to note that the subgroup of 5,014, used in the secondary analysis of factors associated with a positive HIV test, was derived from the total number of individuals recruited during both observation phases (n=7,102). Out of these, 2,074 were not tested for HIV due to either a previously known HIV diagnosis or refusal to test, and 14 had undetermined HIV test result, as illustrated in Figure 2 and clarified in the Results section at line 331. This led to a final count of 5,014 participants being included in the analysis of factors associated with a positive HIV test result. The characteristics of this cohort are displayed in the “Total” column of Table 3. 

• Results, page 45, line 361: the coauthors state targeted PITC would have identified 72% of all persons with a positive result (306 of 509), but it is not clear if most of these were picked up by the healthcare workers who referred them for PITC despite not meeting the algorithm criteria? 

RE: Thank you for the comment. As illustrated in Figure 2, out of the 509 individuals with a positive HIV test, 364 were eligible for HIV testing according to the current targeted PITC algorithm, with healthcare providers referring 360 of these for testing. Of the 145 individuals with a positive HIV test but not eligible under the current targeted PITC criteria, 116 were also referred for testing. This reflects a practice during the observation phases that leaned more towards universal rather than targeted PITC, likely influenced by the Hawthorne effect. However, it is important to note that at line 366, we aimed to describe the number of HIV diagnoses missed by the current targeted PITC algorithm eligibility, regardless of healthcare provider referrals. We have provided additional clarification at this line. The performance of targeted PITC as executed by healthcare providers, measured as HIV test positivity and being the primary outcome of this study, has been thoroughly presented earlier in the section of HIV positivity yield in the Results at line 301. 

Figure 2 is very unclear - impossible to read.

RE: Thank you for the comment, and we apologise for any confusion caused. As previously mentioned, in response to feedback during the first revision, we improved the resolution of Figure 2, an extensive flow diagram. Unfortunately, despite our efforts to enhance its resolution, it seems the clarity remained compromised in the PDF document generated by the submission process. We believe that the figure can be downloaded from the submission system in its original format for clearer viewing. To aid the review process, and as the reviewer must have already noted, we have attached Figure 2 to this letter.

Thank you for considering this article for publication in your journal. I look forward to hearing from you soon.

Please accept my highest assurances.

Yours sincerely,

Anna Saura Lázaro

---

## [Editor Report · Decision Letter 2]

19 Apr 2024

Enhancing HIV positivity yield in southern Mozambique: the effect of a Ministry of Health training module in targeted provider-initiated testing and counselling

PONE-D-23-28399R2

Dear Dr. Saura-Lázaro,

We’re pleased to inform you that your manuscript has been judged scientifically suitable for publication and will be formally accepted for publication once it meets all outstanding technical requirements.

Kind regards,

Hamufare Dumisani Dumisani Mugauri, Ph.D. Public Health

Academic Editor

PLOS ONE
---

## [Editor Report · Acceptance letter]

29 Apr 2024

PONE-D-23-28399R2 

PLOS ONE

Dear Dr. Saura-Lázaro, 

I'm pleased to inform you that your manuscript has been deemed suitable for publication in PLOS ONE. Congratulations! Your manuscript is now being handed over to our production team.

Kind regards, 

on behalf of

Mr Hamufare Dumisani Dumisani Mugauri 

Academic Editor

PLOS ONE